# Transparent Packaging for Tea: Exploring the Role of Trust in Emerging Markets for Tea Beverages

**DOI:** 10.3390/foods14223893

**Published:** 2025-11-14

**Authors:** Yinqiang Yu, Chitin Hon, Yulin Sun, Ying-Tuan Lo, Pengfei Cheng, Lei Zheng, Charles Spence

**Affiliations:** 1The Institute for Sustainable Development, Macau University of Science and Technology, Taipa 999078, Macao; 2School of Business, Macau University of Science and Technology, Taipa 999078, Macao; 3Faculty of Innovation Engineering, Macau University of Science and Technology, Taipa 999078, Macao; 4Entrepreneurship and Enterprise Hub, Xi’an Jiaotong–Liverpool University, Suzhou 215123, China; 5Faculty of Medicine, Macau University of Science and Technology, Taipa 999078, Macao; 6Crossmodal Research Laboratory, Department of Experimental Psychology, University of Oxford, Oxford OX1 3PS, UK

**Keywords:** health marketing, food marketing, tea beverage packaging, packaging transparency

## Abstract

Tea beverages, which combine the health benefits of traditional tea with convenient consumption, are experiencing rapid market growth. Despite the widespread use of transparent packaging and diverse colour schemes, the influence of such packaging strategies on consumer perception and purchase intention remains poorly understood. The present research therefore aimed to investigate the impact of transparent packaging on consumers’ willingness to purchase tea beverages, as well as the underlying psychological processes, based on trust theory. Across three experimental studies, the results showed that consumers reported a higher purchase intention for tea beverages with transparent (vs. opaque) packaging (Studies 1–3), regardless of whether it was green tea or black tea (Study 2), or a well-known or lesser-known brand (Study 3). Moreover, trust mediated the relationship between transparent (vs. opaque) packaging and purchase intention (Studies 2 and 3). Furthermore, brand awareness moderated the effect of transparent (vs. opaque) packaging on trust and purchase intentions, with stronger effects observed for well-known brands as compared to lesser-known ones (Study 3). In addition, for opaque packaging, green (vs. black) tea beverages with green (vs. red) colour packaging were perceived as more trustworthy, resulting in higher purchase intentions (Studies 1 and 2). These findings identify transparent packaging as a significant marketing tool that can help to increase consumer trust while at the same time promoting positive evaluations and purchase intentions. These insights provide strategic implications for brands operating in the rapidly growing tea beverage market.

## 1. Introduction

Tea beverages are made from tea leaf extracts that preserve the health benefits of traditional tea while offering convenience for consumption [1,2]. Tea drinks are generally classified as white, yellow, green, oolong, black, or dark tea, based on the degree of oxidation the tea leaves undergo during processing. In general, tea drinks contain polyphenols, catechins, amino acids, and vitamins, which are considered beneficial for regulating blood pressure, cholesterol, and blood sugar, as well as for reducing the risk of heart disease [3]. With the growing demand for both health and convenience among young consumers, ready-to-drink (RTD) tea beverages have driven rapid expansion in the beverage market [2,4]. The RTD has become increasingly popular, as it allows for the addition of various ingredients (e.g., sugar) to modify flavour and better meet consumer preferences [5]. According to Statista, the global tea beverage market is projected to reach USD 130 billion by 2029 [6], demonstrating its significant market potential.

Traditionally, tea was generally presented in solid form and had to be brewed prior to consumption. Unlike hot-brewed tea, cold-brewed tea typically has lower bitterness and astringency, making it increasingly popular among young consumers [7]. Today, tea beverages are commonly available in liquid form, and are packaged in both transparent and opaque containers. Transparent packaging enables the consumer to view the product directly, which may enhance their perception of its freshness and overall quality [8,9]. Actually, foods presented in transparent packaging are often considered visually appealing and are associated with higher perceived trustworthiness [8,10,11]. However, consumers often reported negative evaluations of transparent packaging, especially in the case of those products that are visually unappealing [12,13]. Given the opposing effects of transparent packaging on food perception and purchase intention [14], the first research objective of the present research was to investigate the effect of transparent versus opaque packaging on consumers’ purchase intention. Furthermore, since transparent packaging is associated with perceptions of ‘openness’ and ‘trustworthiness’ [8], by adopting trust theory, the second research objective was to explore the psychological mechanisms underlying such relationship.

## 2. Literature Review

### 2.1. Product Packaging Transparency and Consumer Perception

Product packaging not only serves as a tool for protecting the contents but also as a powerful sensory cue that influences food perception and purchase intentions [11,15,16]. Transparent packaging allows consumers to directly view the product inside and assess its quality, which are considered essential factors in food choice [8,9]. In particular, transparent packaging highlights fundamental food characteristics such as colour, texture, and shape, leading consumers to perceive the food as being of higher quality [8,10,17], as well as healthier [18]. According to cue utilization theory, both intrinsic (e.g., material) and extrinsic cues (e.g., package) play a significant role in shaping individuals’ perceptions and behaviours [19,20,21]. Intriguingly, when intrinsic and extrinsic cues match (i.e., when they are consistent), consumers are more likely to form positive evaluations. In contrast, inconsistent cues often make it difficult for consumers to accurately assess the product, potentially resulting in negative evaluations [22,23].

The effect of transparent packaging on food perception varies as a function of the product category [10]. For example, cookies packaged in transparent (vs. opaque) packaging were perceived as crispier, sweeter, and tastier, but less healthy [24]. However, when espresso and Americano coffees were served in transparent glass mugs (vs. white ceramic mugs), consumers expected them to be more aromatic, bitter, hot, and intense [25]. Van Doorn et al. (2014) reported that serving a café latte in a white mug (versus a transparent glass mug) led to expectations of reduced sweetness [26]. This is consistent with crossmodal theory, which posits that the colour of packaging can influence consumers’ expectations about the likely taste or flavour characteristics of a product [16,27,28,29]. For example, sweetness is commonly associated with red and pink, whereas sourness is linked with green and yellow hues [28]. In tea products, colour is associated with perceived bitterness and astringency, with the flavour of green tea typically linked to greenish colour, while black tea is associated with warmer hues [30].

Regarding purchase intention, transparent packaging exhibits two opposing effects: the salience effect, which enhances eye appeal and increases purchase intention, and the monitoring effect, which allows consumers to critically evaluate the product’s quality, potentially decreasing their purchase intention [14]. In particular, Simmonds et al. (2018) found that, across four food categories (e.g., granola, boxed chocolate, dried pasta, and fresh fish), transparent packaging consistently elicited a higher purchase intention [10]. Similarly, organic food in transparent (vs. opaque) packaging is perceived to have higher green perceived value and greater trustworthiness, leading consumers to report stronger purchase intentions [31,32]. Although many studies have demonstrated the positive effects of transparent packaging on purchase intention [8,10,14,33,34], it has also has been shown to exert a negative impact on people’s food choices under certain conditions [8,12,13]. For example, transparent (vs. opaque) packaging was perceived as more instrumental, less esthetic, and less symbolic of quality, resulting in a 30% decrease in sales in one study [13]. The negative impact is especially pronounced when the food elicits a sense of disgust or comes into contact with other disgust-inducing products. However, it has been reported that opaque packaging effectively blocks the contagion effects associated with such disgust-inducing products [12].

Tea is well-known not only for its unique aroma and flavour but also, increasingly, for its well-documented health benefits [1,35]. Transparent materials allow consumers to see the contents, thus enhancing perceived fluency toward the product [36,37]. In particular, an eye-tracking study found that orange juice packaged in transparent glass and plastic bottles attracted more visual fixations and elicited stronger purchase intentions, as compared to packaging in opaque pouches and tetrapacks [38]. In line with that, transparent packaging is believed to enhance expected tastiness and perceived healthiness, thus leading to increased purchase intention [10,18]. Actually, both expected tastiness and perceived healthiness play a crucial role in purchase decision-making [1,35,39,40]. Transparent packaging allows consumers to form expectations about the flavour and healthiness of tea beverages, which may further enhance their purchase intentions. Thus,

**H1:** 
*Tea beverages using transparent (vs. opaque) packaging led consumers to report stronger purchase intention.*


### 2.2. Trust and Brand Awareness

Trust plays an important role in consumers’ purchase intentions [41,42]. Trust is the confidence or expectation that one person has in another to act as anticipated, based on their positive relationship [43]. Trust is widely recognized as a multidimensional construct, commonly divided into two components: cognitive trust, which is based on rational beliefs about a product’s competence or quality, and affective trust, which reflects emotional security and a sense of emotional closeness or attachment [44,45].

Specifically, when consumers trust the food products, they are more likely to purchase them [46]. This is because trust mitigates consumers’ perception of purchase risk, thus leading to stronger purchase intentions [47]. However, when firms exaggerate the environmental performance of their products, consumers perceive the firm as less trustworthy and are therefore less likely to purchase its products [48].

Since transparent packaging is considered to convey notions of “openness” and “trustworthiness”, those food and beverages products presented in transparent packaging are thought to enhance the consumer’ trust [8]. Specifically, trust is regarded as a mediator in the relationship between transparent packaging and positive expectations regarding product quality [10,49]. For example, consumers perceive transparent packaging for juice and chocolate bars to be more trustworthy than opaque packaging, thus leading to stronger purchase intentions for those products with transparent packaging [50]. By allowing consumers to visually inspect the quality, ingredients, and contents of the food, transparent packaging reduces uncertainty, fosters trust, and ultimately encourages purchase decisions [32]. Therefore, incorporating transparent design elements into product packaging is widely regarded as a strategy that positively affects consumer evaluations and purchase intentions [8]. Thus,

**H2:** *Trust mediated the relationship between transparent (vs. opaque) packaging and purchase intention toward tea beverages*.

When consumers purchase beverages, they often rely on the brand to inform their decision. Notably, the effect of transparent packaging on trust is more pronounced for those products that consumers are unfamiliar with, whereas it has less of an impact for products from well-known brands [8]. These familiar brands are often easily recalled and recognized, a concept known as brand awareness, which reflects their prominence in consumers’ minds [51]. Importantly, brand awareness is regarded as a key antecedent of trust [52,53], and has been found to strengthen trust through brand reputation [54,55]. However, consumers tend to distrust unfamiliar products [56], but transparent packaging enhances trust in these products [8]. Therefore,

**H3:** *Brand awareness moderated the effect of transparent (vs. opaque) packaging on trust and purchase intention. Specifically, transparent (vs. opaque) packaging increased trust and purchase intention for products with low brand awareness, but not for those with high brand awareness*.

### 2.3. The Present Study

Based on crossmodal research [57] and trust theory [58,59], three studies were conducted to examine whether and how transparent (vs. opaque) packaging impacts consumers’ purchase intention for tea beverages. In Study 1, the direct effect of transparent (vs. opaque) packaging on purchase intention was investigated. Study 2 extended this investigation by exploring the impact of packaging transparency on the intention to purchase for both green tea and black tea—the two most widely consumed types globally—and further examined the mediating role of trust in this relationship. Moreover, since consumers may consider brand reputation when purchasing tea beverages, Study 3 explored the moderating role of brand awareness on the relationship between transparent (vs. opaque) packaging and trust, and between transparent (vs. opaque) packaging and purchase intention. Lastly, given the variation in the colour of opaque packaging, and drawing on the colour–flavour congruency effect [16,28], our research investigated how the colour of opaque packaging impacts consumers’ perceptions and purchase intention (see Figure 1).

## 3. Study 1: The Effect of Transparent Packaging on the Purchase Intention

Study 1 aimed to examine the effect of transparent (vs. opaque) packaging on consumers’ purchase intention. Since opaque packaging for tea beverages often comes in different colours, this study focused on three common colours: Green, yellow, and white. These colours are frequently used in the market, representing the colour of tea leaves (green), the colour of brewed tea (yellow), and a neutral background (white).

### 3.1. Participants

Based on prior research on packaging transparency [13], power analysis was conducted using G*Power 3.1.9.7. The analysis indicated that a minimum sample size of 152 participants was required to detect an effect size of *f* = 0.31 with a statistical power of (1 − β) > 0.90. To account for potential exclusions due to failed attention checks, a total of two hundred Chinese participants were recruited from Credamo (www.credamo.com) in December 2024. Six participants were removed due to their failure to pass the attention check, leaving a final sample of 194 (*M_age_* = 33.68 ± 13.71 years; 64.95% Women). The participants were randomly assigned to one of four groups (*n* = 44 for transparent packaging group, *n* = 50 per opaque packaging group) and exposed to tea beverages in either transparent or opaque packaging in green, yellow, or white.

### 3.2. Procedure

Participants were instructed to imagine the following scenario: “You enter a supermarket to buy a beverage and notice a locally branded tea beverage. The tea beverage is packaged using [(a) transparent packaging/(b) opaque white packaging/(c) opaque green packaging/(d) opaque yellow packaging].” After reading the scenario and viewing the corresponding image of the product, participants completed a series of study measures (see details in Appendix A), including purchase intention, perceived healthiness, and perceived tastiness.

### 3.3. Manipulation Check

The manipulation check was successful (
$\chi_{\left(1\right)}^{2}$ = 194, *p* < 0.001). Participants were asked: ‘What kind of packaging was used for the tea beverage?’ (1 = Transparent, 2 = Opaque).

### 3.4. Results

The results showed that there was a significant main effect of transparent packaging on purchase intention (*F*[3, 190] = 10.51, *p* < 0.001,
$\eta_{p}^{2}$ = 0.14; see Figure 2A). Specifically, participants showed a higher purchase intention for tea beverages with transparent packaging (*M =* 3.43 ± 1.25) than those with green (*M =* 2.93 ± 1.16; *t =* 2.02, *p* = 0.046, Cohen’s d *=* 0.42), yellow (*M =* 2.42 ± 1.21; *t =* 3.98, *p* < 0.001, Cohen’s d *=* 0.82), or white opaque packaging (*M =* 2.21 ± 0.99; *t =* 5.29, *p* < 0.001, Cohen’s d *=* 1.09). Interestingly, purchase intention also differed among the opaque packaging colours. Participants reported a higher purchase intention for tea beverages with green opaque packaging than for those with yellow (*t =* 2.15, *p* = 0.034, Cohen’s d *=* 0.43) or white opaque packaging (*t =* 3.34, *p* = 0.001, Cohen’s d *=* 0.69). However, there was no significant difference in purchase intention between the yellow and white packaging (*t =* 0.95, *p* = 0.345, Cohen’s d *=* 0.19).

Additionally, transparent (vs. opaque) packaging positively affected perceived healthiness (*F*[3, 190] = 5.86, *p* < 0.001,
$\eta_{p}^{2}$ = 0.09; see Figure 2B), but had no significant effect on perceived tastiness (*F*[3, 190] = 1.99, *p* = 0.118,
$\eta_{p}^{2}$ = 0.03; see Figure 2C). Specifically, participants reported higher perceived healthiness for the tea beverage with transparent packaging (*M =* 3.73 ± 1.10) as compared to those with green (*M =*3.31 ± 1.07; *t =* 1.88, *p* = 0.064, Cohen’s d *=* 0.39), yellow (*M =*2.92 ± 1.22; *t =* 3.35, *p* = 0.001, Cohen’s d *=* 0.69), or white (*M =* 2.90 ± 0.99; *t =* 3.83, *p* < 0.001, Cohen’s d *=* 0.79) opaque packaging. However, opaque packaging colour (i.e., green, yellow, white, *n* = 50 per opaque packaging group) had a nonsignificant effect on perceived tastiness (*F*[2, 147] = 2.18, *p* = 0.117,
$\eta_{p}^{2}$ = 0.03).

### 3.5. Discussion

The results of Study 1 show that transparent packaging consistently enhances consumers’ purchase intentions and perceived healthiness of tea beverages, compared to opaque packaging. In particular, tea beverages that were in transparent packaging led to a significantly higher purchase intention (*t =* 4.51, *p* < 0.001, Cohen’s d = 0.68) and perceived healthiness (*t =* 3.62, *p* < 0.001, Cohen’s d = 0.55) than the midpoint of the scale (i.e., ‘unsure’). Interestingly, green opaque packaging elicited a stronger purchase intention than the other two opaque packaging colours, but this difference was not statistically significant when compared to the midpoint of the scale (*t =* −0.43, *p* = 0.672, Cohen’s d = 0.06). However, there were nonsignificant differences in perceived healthiness and tastiness amongst the opaque coloured packaging. These results therefore highlight the importance of transparency in shaping consumers’ perceptions of product quality and healthiness.

## 4. Study 2: The Effect of Type of Tea and Packaging Transparency on Consumer Preference for Tea Beverages

In Study 2, we aimed to explore whether the effect of transparent (vs. opaque) packaging on tea beverages varies across as a function of the type of tea. Given the prevalence of green and black tea consumption, particularly in East Asia, and their respective colour associations (green for green tea and red—as it is traditionally termed in China—for black tea), opaque green and red packaging were used. Thus, we used a mixed-subject design with tea type as the between-subject factor (green tea vs. black tea) and packaging type as the within-subject factor (transparent, opaque green, opaque red).

### 4.1. Participants

Based on the main effect of transparent packaging on purchase intention (*f*^2^ = 0.16), a power analysis (statistical power = 0.90, α = 0.05) for a mixed-subjects design and logistic regression estimated a minimum sample size of 83 participants for Study 2. Thus, we recruited one hundred and twenty Chinese participants (*M_age_* = 25.17 ± 5.88 years; 60% women) from Credamo (https://www.credamo.com (accessed on 11 October 2025)) during January 2025. The participants were randomly assigned to one of two groups (*n* = 60 for green tea and 60 for black tea).

### 4.2. Procedure

Participants were instructed to imagine the following scenario: “You enter a supermarket to buy a beverage and notice a locally branded tea beverage. Three types of tea beverages were presented on the shelf, each having a different packaging style: transparent, opaque green, or opaque red.” The positions of the product images were counterbalanced to control for position effects. They were then asked to rank the four tea beverages based on their purchase intention, with the product ranked first considered the one they would choose to purchase. Finally, participants reported trust, perceived healthiness, and perceived tastiness (See details in Appendix A).

### 4.3. Results

Our results showed a significant interaction effect of packaging type and tea type on the consumer’s preference for tea beverages (
$\chi_{\left(2\right)}^{2}$ = 13.51, *p* = 0.001, see Figure 3A). In particular, consumers tended to choose transparent packaging for both green tea (*n* = 43, 71.7%) and black tea (*n* = 47, 78.3%). However, consumers chose the green tea with opaque green packaging (*n* = 15, 25.0%) more often than the opaque red packaging (*n* = 2, 3.3%), and they chose the black tea with opaque red packaging (*n* = 10, 16.7%) more often than the opaque green packaging (*n* = 3, 5.0%).

Furthermore, the analysis of the results highlighted a significant interaction of packaging type and tea type on trust (*F*[2, 236] = 10.27, *p* < 0.001,
$\eta_{p}^{2}$ = 0.08; see Figure 3B). Participants reported greater trust in tea beverages with transparent packaging for both the green (*M =* 4.26 ± 0.72) and black tea (*M =* 4.14 ± 0.72), compared to those with opaque red packaging (Green tea: *M =* 2.97 ± 0.80, *t =* 2.02, *p* < 0.001, Cohen’s d *=* 1.28; Black tea: *M =* 3.54 ± 0.98, *t =* 4.56, *p* < 0.001, Cohen’s d *=* 0.59) and opaque green packaging (Green tea: *M =* 3.62 ± 1.00, *t =* 4.53, *p* < 0.001, Cohen’s d *=* 0.59; Black tea: *M =* 3.29 ± 0.94, *t =* 5.50, *p* < 0.001, Cohen’s d *=* 0.71). Notably, green tea in opaque green packaging was perceived as more trustworthy than when presented in opaque red packaging (*t =* 4.59, *p* < 0.001, Cohen’s d *=* 0.59), but no such effect was observed for black tea (*t =* 1.41, *p* = 0.163, Cohen’s d *=* 0.182).

Additionally, our research also examined the effect of packaging type on perceived tastiness and perceived healthiness for both green and black tea. The results revealed an interaction effect of transparent packaging (compared to opaque red and opaque green packaging) and tea type on both perceived tastiness (*F*[2, 236] = 16.01, *p* < 0.001,
$\eta_{p}^{2}$ = 0.12; see Figure 3C) and perceived healthiness (*F*[2, 236] = 10.79, *p* < 0.001,
$\eta_{p}^{2}$ = 0.08; see Figure 3D). In particular, the participants expected tea beverages packaged in transparent containers to be tastier, for both green tea (*M =* 3.93 ± 0.88) and black tea (*M =* 3.83 ± 0.75), as compared to those in opaque red packaging (Green tea: *M =* 2.76 ± 0.72, *t =* 8.08, *p* < 0.001, Cohen’s d *=* 1.04; Black tea: *M =* 3.50 ± 0.86, *t =* 2.35, *p* = 0.022, Cohen’s d *=* 0.30), and opaque green packaging (Green tea: *M =* 3.43 ± 0.89, *t =* 3.87, *p* < 0.001, Cohen’s d *=* 0.50; Black tea: *M =* 3.09 ± 0.88, *t =* 4.81, *p* < 0.001, Cohen’s d *=* 0.62). Furthermore, participants expected the green tea beverage to be tastier when presented in green than in red packaging (*t =* 5.36, *p* < 0.001, Cohen’s d *=* 0.69). In contrast, they expected the black tea beverage to be tastier in red than in green packaging (*t =* 2.63, *p* = 0.011, Cohen’s d *=* 0.34).

In terms of perceived healthiness, participants considered tea beverages in transparent packaging to be healthier for both green tea (*M =* 4.13 ± 0.73) and black tea (*M =* 4.03 ± 0.90), compared to the one with opaque red packaging (Green tea: *M =* 2.74 ± 0.74, *t =* 10.27, *p* < 0.001, Cohen’s d *=* 1.33; Black tea: *M =* 3.32 ± 1.03, *t =* 5.88, *p* < 0.001, Cohen’s d *=* 0.76) and opaque green packaging (Green tea: *M =* 3.53 ± 1.01, *t =* 4.49, *p* < 0.001, Cohen’s d *=* 0.58; Black tea: *M =* 3.27 ± 0.93; *t =* 5.72, *p* < 0.001, Cohen’s d *=* 0.74). Participants rated green tea in opaque green containers as looking healthier than green tea in opaque red containers (*t =* 6.19, *p* < 0.001, Cohen’s d *=* 0.80). However, there was no significant difference between the opaque red and opaque green packaging for black tea (*t =* 0.22, *p* = 0.746, Cohen’s d *=* 0.04).

Since participants considered tea beverages with transparent packaging to be more trustworthy, palatable, and healthier, as compared to opaque packaging (regardless of colour), the two types of opaque packaging (red and green) were combined into a single opaque packaging category. A moderated mediation model was constructed using the PROCESS Model 7 (5000 resamples), where packaging type (transparent vs. opaque) was the predictor, trust was the mediator, tea type (green vs. black) was the moderator, and purchase intention was the outcome. The results showed that the mediating effect was significant for both green tea (*Effect =* 1.12, s.e. = 0.28, *p* < 0.001, 95% CI [0.66, 1.79]) as well as for black tea (*Effect =* 0.84, s.e. = 0.25, *p* < 0.001, 95% CI [0.45, 1.40]). However, the difference in the moderated mediation effect was nonsignificant (*Effect =* 0.28, s.e. = 0.22, *p* = 0.203, 95% CI [−0.13, 0.73]).

### 4.4. Discussion

The results of Study 2 once again demonstrated that participants were more likely to choose tea beverages with transparent packaging than those with opaque packaging, regardless of whether the product was green tea or black tea. Trust was identified as a significant mediator in the relationship between packaging type and beverage choice. Specifically, participants reported higher levels of trust in tea beverages presented in transparent packaging, which in turn increased their preference for those products. This mediating effect of trust was consistent across both green tea and black tea conditions. Additionally, tea beverages presented in transparent packaging were expected to be tastier and healthier than those in opaque packaging, even when the latter’s colour matched the tea type (e.g., green for green tea, red for black tea). Our results suggest that transparent packaging enhances consumers’ trust in tea beverages, which in turn leads to higher purchase intentions and more positive product evaluations, including greater perceived healthiness and tastiness, compared to opaque packaging.

## 5. Study 3: The Moderating Role of Brand Awareness

The aim of Study 3 was to examine whether brand awareness moderates the effect of transparent packaging on tea beverage preferences. To achieve this, we used a mixed-subjects design, with brand awareness as the between-subjects factor (Well-known vs. Lesser-known) and packaging type as the within-subjects factor (Transparent, Opaque white, Opaque green). Two brands were selected: the well-known brand, Suntory, whose annual sales reached CNY 104.6 billion in 2022; and the lesser-known brand, Tongzhen, which is only sold in limited regions in China.

### 5.1. Participants

Based on the interaction effect of packaging type and tea type on preference (*w* = 0.34), the power analysis (power = 0.90, α = 0.05) estimated a minimum sample size of 110 participants. Thus, a total of 120 Chinese participants were recruited from Credamo in April 2025. After excluding 4 participants who failed the attention check, 116 participants (M_age_ = 25.09 ± 5.06 years; 72.4% women) were retained for further data analysis: 58 in the well-known brand group and 58 in the lesser-known brand group.

### 5.2. Procedure

They were instructed to imagine the following scenario: “You enter a supermarket to buy a beverage and notice a brand of tea beverage. There are three types of this tea beverage on the shelf, each presented with a different packaging style: transparent, opaque white, or opaque green.” The positions of the product images were counterbalanced to control for potential position effects. Participants were then asked to choose one tea beverage from the shelf and were informed that they would complete the same study measures as in Study 2.

### 5.3. Manipulation Check

The brand awareness manipulation check (‘How popular do you think this tea brand is?’, 1 = Well-known, 2 = Lesser-known). was successful (
$\chi_{\left(1\right)}^{2}$ = 116, *p* < 0.001).

### 5.4. Results

Our research revealed a significant main effect of packaging type, with most consumers preferring transparent packaging (*n* = 87,
$\chi_{\left(2\right)}^{2}$ = 118.79, *p* < 0.001) over opaque green (*n* = 13) and opaque white (*n* = 16) packaging. Moreover, brand awareness moderated the effect of packaging type on consumer preferences (
$\chi_{\left(2\right)}^{2}$ = 6.16, *p* = 0.046, see Figure 4A), with a stronger effect observed for the well-known brand (Transparent: *n* = 49; Opaque green: *n* = 3; Opaque white: *n* = 6) compared to the less familiar brand (Transparent: *n* = 38; Opaque green: *n* = 10; Opaque white: *n* = 10). As there was no difference between opaque green and opaque white in consumer preferences, the two packaging types were combined for further data analysis, resulting in packaging type being categorized into two levels: transparent and opaque.

Furthermore, the results of Experiment 3 revealed that brand awareness moderated the effect of transparent (vs. opaque) packaging on trust (*F*[1, 114] = 4.64, *p* = 0.033,
$\eta_{p}^{2}$ = 0.04; see Figure 4B). Participants reported greater trust in tea beverages with transparent packaging for the well-known brand (M = 4.62 ± 0.50) as compared to those with opaque packaging (*M =* 3.41 ± 0.47, *t =* 12.17, *p* < 0.001, Cohen’s d *=* 1.60). This effect, though significant, was smaller for the lesser-known brand, with transparent packaging (M = 4.06 ± 0.59) eliciting greater trust than opaque packaging (*M =* 3.16 ± 0.71, *t =* 8.65, *p* < 0.001, Cohen’s d *=* 1.14).

Additionally, a moderated mediation model was conducted using the PROCESS Model 7 (5000 resamples), where transparent (vs. opaque) packaging was the predictor, trust was the mediator, brand awareness (well-known vs. lesser-known) was the moderator, and tea beverage choice was the outcome. The results highlighted a significant moderated mediating effect (*Effect =* 0.26, s.e. = 0.17, *p* = 0.126, 95% CI [0.01, 0.68]; see Figure 5). Particularly, the mediating effect of trust was stronger for the well-known brand Suntory (*Effect =* 1.06, s.e. = 0.40, *p* = 0.008, 95% CI [0.33, 1.93]), than for the lesser-known brand (*Effect =* 0.80, s.e. = 0.30, *p* = 0.008, 95% CI [0.25, 1.45]).

We further explored the effect of packaging type on perceived tastiness and healthiness for both green and black tea. The results revealed an interaction between brand awareness and transparent packaging on perceived tastiness (*F*[1, 114] = 5.46, *p* = 0.021,
$\eta_{p}^{2}$ = 0.05) and healthiness (*F*[1, 114] = 9.61, *p* = 0.002,
$\eta_{p}^{2}$ = 0.08). Participants rated tea beverages in transparent containers as looking tastier, with a stronger effect observed for the lesser-known brand (Transparent: *M =* 3.36 ± 0.72; Opaque: *M =* 2.85± 0.58; *t =* 4.81, *p* < 0.001, Cohen’s d *=* 0.63), while the effect was nonsignificant for the well-known brand (Transparent: *M =* 3.71 ± 1.00; Opaque: *M =* 3.56 ± 0.73; *t =* 1.27, *p* = 0.208, Cohen’s d *=* 0.17). In addition, participants also perceived tea beverages in transparent packaging as healthier, with a stronger effect observed for the well-known brand (Transparent: *M =* 4.28 ± 0.60; Opaque: *M =* 3.22 ± 0.49; *t =* 10.70, *p* < 0.001, Cohen’s d *=* 1.40), but a weaker effect for the lesser-known brand (Transparent: *M =* 3.88 ± 0.70; Opaque: *M =* 3.28 ± 0.71; *t =* 5.69, *p* < 0.001, Cohen’s d *=* 0.75).

### 5.5. Discussion

In Study 3, using two real brands, we replicated the finding that consumers tended to choose tea beverages with transparent packaging rather than those with opaque packaging. Furthermore, consumers perceived tea beverages in transparent packaging to be more trustworthy, appealing, and healthier. Notably, a moderating effect of brand awareness on this preference was identified. The moderated mediation model revealed that trust served as the mediator in the relationship between packaging type (transparent vs. opaque) and brand awareness for both well-known and lesser-known brands. These findings therefore suggest that both lesser-known and well-established brands can benefit from adopting transparent packaging. By enhancing consumer trust, transparent packaging has the potential to positively influence purchase intentions and ultimately boost sales.

## 6. General Discussion

Our research revealed that, compared to opaque packaging, transparent packaging enhanced consumer purchase intentions and fostered more favourable perceptions of tea beverages, particularly regarding perceived healthiness and tastiness. Moreover, trust mediated the relationship between transparent (vs. opaque) packaging and purchase intention, regardless of whether the beverage was green tea or black tea. Additionally, brand awareness significantly moderated this mediation effect. In particular, consumers reported greater trust in well-known brands, which further strengthened their intention to purchase tea beverages from them.

### 6.1. Findings

In this research, consumers reported higher trust and stronger purchase intention for tea beverages with transparent (vs. opaque) packaging. Although transparent packaging is thought to have opposing effects on purchase intention [14], this research demonstrates its importance for tea beverages in helping to build customer–product trust as well as enhancing consumer purchase intention. Notably, this effect was consistent across both green and black teas, as well as for well-known and unfamiliar brands. Since flavour and health benefits are crucial factors in purchasing tea beverages [1,35], consumers often evaluate tea qualities based on the colour, clarity, and darkness of the beverage. Thus, allowing consumers to see the product inside the packaging facilitates quality assessment [17,36], thus leading to increased purchase intention [10,18]. Moreover, our results support the colour–flavour congruency effect [16,28], indicating that consumers rate green tea as best packaged in green and black tea (‘red tea’ in Chinese) as best packaged in red [30]. According to cue utilization theory, individuals experience difficulty processing inconsistent cues, which leads to negative consumer evaluations and reduced purchase intentions [22,23]. In line with that, the pairing of opaque green packaging with black tea or opaque red packaging with green tea represented inconsistent cues, which led to significantly lower purchase intentions. In contrast, consistent cues facilitate processing fluency, leading consumers to evaluate the tea beverages more positively [23,36]. However, disconfirmation of these expectations resulted in the lowest purchase intentions and poorer perceptions of food quality, such as taste and perceived healthiness, aligning with the results of previous research [16,60]. Notably, in Chinese culture, red is traditionally regarded as a symbol of good luck [61]. During Chinese festivals, for instance, black tea presented in red opaque packaging may evoke stronger consumer evaluations, thus enhancing perceived value and purchase intention. However, this research also demonstrated that green tea in green opaque packaging received more favourable evaluations than green tea in red packaging, suggesting a colour–flavour congruency effect in opaque packaging for tea beverages.

Moreover, our findings that transparent packaging enhanced trust highlight the association between ‘transparency’ and trust, often conveyed through the notion of ‘having nothing to hide’ [8]. In line with our findings, transparent packaging is believed to increase consumers’ trust in the quality and reliability of the product because it allows consumers to see the contents inside [62]. Moreover, transparency is semantically associated with trustworthiness reinforcing consumers’ perceptions of product credibility and integrity (e.g., free from artificial additives) [8,63,64]. Thus, consumers are more likely to exhibit favourable evaluations toward products that are perceived as credible and trustworthy [19,20,23]. More importantly, transparency reduces perceived uncertainty regarding the product’s quality and safety, thus strengthening consumer trust and boosting consumers’ purchase intention [32,34]. Previous research has found that, for those products with transparent packaging, consumers tend to place greater trust in unfamiliar brands and perceive their product quality to be higher compared to familiar brands [8]. However, the research reported here revealed that transparent packaging boosted trust not only for unfamiliar brands but also for well-known ones. In fact, higher brand awareness and familiarity with a well-known brand tends to produce a halo effect, leading consumers to evaluate its products more favourably [65]. In contrast, lesser-known brands are unlikely to have highly established brand equity which is necessary for ensuring that their reputation is stable and has less excess uncertainty among consumers regarding product quality [66].

Another plausible explanation for the stronger mediating effect of trust observed in well-known (vs. lesser-known) brands is the transferability of trust [67]. It perhaps enables consumers to more readily construct trust in transparent packaging when it is associated with well-known brands. According to trust transfer theory, consumers tend to exhibit greater trust in a product when it is associated with a trustworthy brand [67,68,69]. For example, many restaurants seek Michelin recognition as a means of enhancing consumer trust in their offerings. Consequently, products from well-known brands may more effortlessly evoke positive perceptions of reliability and quality, stemming from consumers’ pre-existing trust in the brand. However, lesser-known tea beverage brands often lack an established foundation of consumer trust, leading individuals to seek additional informational cues to form trust judgments. Consequently, compared to well-known brands, the impact of transparent packaging on consumer trust is more limited for lesser-known brands.

Based on our findings, even consumers’ trust toward unfamiliar brands with superior transparency was lower as compared to familiar brands whose packaging is transparent; however, transparency of lesser-known brands in-turn enhances their level of credibility in comparison to opaque options. Thus, when tea beverage brands have low brand awareness, the adoption of transparent packaging is a good approach to build consumer trust which in turn would lead to higher sales. Since products from well-known brands are often priced higher than those from lesser-known brands and are generally perceived to be of higher quality, this may further enhance consumers’ purchase intentions.

When tea beverages were presented in transparent packaging, they were rated as looking healthier than those in opaque packaging. Previous research has reported conflicting findings regarding the impact of transparent packaging on the perceived healthiness of food and beverages [8,18,24,70]. One possible explanation relates to the particular contents that happen to be visible through the packaging. Since tea is generally regarded as ‘healthy’, ‘refreshing’, and ‘natural’ [1,4,35], seeing the tea inside the packaging aligns with consumers’ expectations, thus enhancing their perception of healthiness [18]. Conversely, when the liquid inside fails to meet consumer expectations, it can lead to lower healthiness evaluations [8,70,71]. Regarding perceived tastiness, our findings were mixed: Studies 2 and 3 showed that transparent packaging increased perceptions of tastiness, while Study 1 found no significant difference. A possible explanation is the halo effect, whereby consumers associate transparent packaging with greater trustworthiness, which then boosts taste expectations. In Study 1, where we referred to tea beverages more generally, consumers may have found it harder to imagine the specific taste. In contrast, Studies 2 and 3 specified green and black teas, in addition to well-known and lesser-known brands, which likely helped consumers form clearer expectations. This process enhances mental simulation of the consumption experience [33] and amplifies hedonic evaluations [24], thus leading consumers to expect that the tea will be tastier.

### 6.2. Practical Implications

Our findings suggest that transparent packaging can serve as an effective marketing tool for tea beverages. By enhancing consumers’ perceptions of trust, healthiness, and tastiness, transparent packaging positively impacts purchase intentions. Therefore, tea beverage companies are encouraged to adopt transparent packaging to increase their product appeal and better meet consumer expectations, especially within health-conscious markets [13]. This approach has the potential to drive higher sales and strengthen brand perception.

Transparent packaging is especially beneficial for brands with low market awareness. Our research revealed that transparent packaging functions as a compensatory strategy to help build initial trust among consumers. Transparent packaging helps reduce uncertainty and reinforces positive perceptions towards food, making it a practical choice for emerging or less familiar tea brands. For well-established brands, transparent packaging also offers value by serving as a visual cue that signals both honesty and openness [8].

Although some brands adopt opaque packaging for their tea beverages to prominently display the brand name, nutritional labels, and other product details, the importance of colour–flavour congruency in packaging design should not be overlooked [16,28]. Use of the ‘appropriate’ colours not only helps prevent any ‘disconfirmation of expectation’ [60], but also increases cognitive fluency [36,37], which is a crucial factor in consumers’ decision-making. For example, the Chinese food brand *Uni-President* predominantly uses red in the opaque part of the packaging for its iced black tea [72], while its green tea products [73], similar to those of the Japanese brand *Ito En* [74], feature large areas of green as the main colour of the opaque packaging. Hence, when firms use opaque packaging for tea beverages, colour–flavour congruency should be considered to effectively influence consumers’ food perceptions. In addition to ensuring colour–flavour congruency, colour contrast is crucial for enhancing the visual appeal [16]. Since tea colour varies in intensity and flavour, future studies should focus on optimizing the contrast between packaging colour and the tea to enhance visibility.

### 6.3. Limitations and Future Research

This research has several limitations. First, our research relied solely on self-reported measures and laboratory experiments, which limit the external validity of our findings [40]. Since contextual factors (e.g., seasonality and the social desirability effect) were not controlled in the present study, future research could incorporate complementary assessments to obtain a more comprehensive understanding of the factors influencing consumers’ purchase intentions as a function of the time of year [75]. For example, Augmented reality and virtual reality technologies could be employed in future research to examine the impact of transparent packaging on consumers’ willingness to purchase tea beverages. Second, while this research merely focused on packaging transparency, it is crucial to recognize that other related factors, such as packaging materials (e.g., PET or glass) [25,26] and surface finish [76] can also influence the consumers’ product perception. As Simmonds and Spence (2019) suggested, packaging that balances visibility of the product with protection against negative effects, such as catechin degradation and chlorophyll photooxidation, can assist product designers in optimizing the overall consumer experience [77]. Given that transparent packaging may increase the susceptibility of polyphenol-rich teas to light-induced oxidation, future research should further examine the chemical and sensory trade-offs associated with packaging design choices. Future research should therefore examine the oxidative risks of transparent packaging in greater depth, particularly in the context of light-sensitive products.

Third, although tea originated in China, a country with a large population of tea drinkers and accounting for over 85% of the global tea supply [78], it is important to note that the colour of tea can vary significantly depending on the tea type, concentration, and processing methods used in different countries. In this study, we used three common opaque colours, green, yellow, and white, to represent tea leaves, brewed tea, and a neutral background, respectively. However, some consumers may perceive the colour of brewed tea as more ‘brownish’, depending on their personal preferences or cultural associations. Future research should explore a wider range of hues or use more precise colour matching to better capture the nuanced effects of packaging colour on consumer perception. Fourth, in the present research, transparency was operationalized as a binary variable (transparent vs. opaque). However, in practice, transparency exists along a continuum, which can be manipulated through varying degrees of translucency or the inclusion of partial window designs. Future research could investigate how incremental variations in packaging transparency influence consumer perceptions and behaviours.

## 7. Conclusions

Across the three studies reported here, transparent (vs. opaque) packaging was consistently shown to lead to more positive consumer responses when choosing tea beverages, as they are perceived to be more trustworthy. This effect was observed for both green tea and black tea, as well as for well-known and lesser-known brands. Moreover, the results reveal that brand awareness moderates the impact of transparent packaging on both trust and purchase intentions, with transparent packaging being particularly effective for well-known brands. Additionally, for opaque packaging, congruence between tea type and packaging colour enhances trust and boosts purchase likelihood—green for green tea and red for black tea.

## Figures and Tables

**Figure 1 foods-14-03893-f001:**
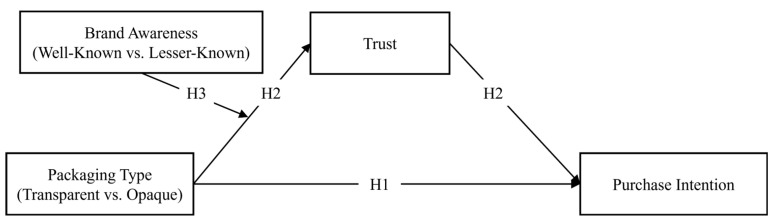
Conceptual model.

**Figure 2 foods-14-03893-f002:**
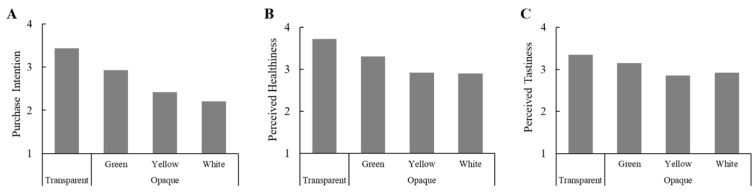
Main effects of transparent packaging on purchase intention, perceived healthiness, and perceived tastiness in Study 1. (**A**) Participants reported higher purchase intention for tea beverages with transparent packaging as compared with green, yellow, or white opaque packaging; (**B**) Tea beverages with transparent packaging were perceived to be healthier than those with opaque packaging; (**C**) The perceived tastiness of tea beverages in transparent and opaque packaging did not differ significantly.

**Figure 3 foods-14-03893-f003:**
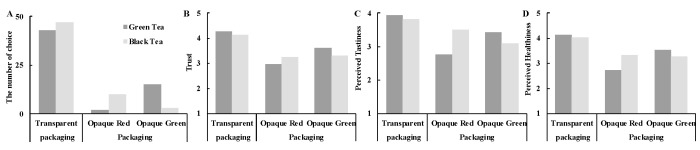
The interaction effect of packaging type (transparent packaging vs. opaque red packaging vs. opaque green packaging) and tea type (green tea vs. black tea) on tea preference, trust, perceived tastiness, and perceived healthiness in Study 2. (**A**) Consumers showed a stronger preference for transparent packaging. Within the opaque packaging conditions, green tea was more preferred in green packaging, whereas black tea was more preferred in red packaging. (**B**) Tea beverages in transparent packaging were rated as more trustworthy. Among opaque packages, green tea in green packaging was perceived as more trustworthy than in red packaging. However, no significant difference was found for black tea between the two opaque colors. (**C**) Tea in transparent packaging was perceived as more tasty. For opaque packaging, green tea was rated tastier in green packaging, and black tea was rated tastier in red packaging. (**D**) Beverages in transparent packaging were rated as healthier. Within the opaque conditions, green tea was perceived as healthier in green packaging than in red, while no difference in healthiness perception was found for black tea.

**Figure 4 foods-14-03893-f004:**
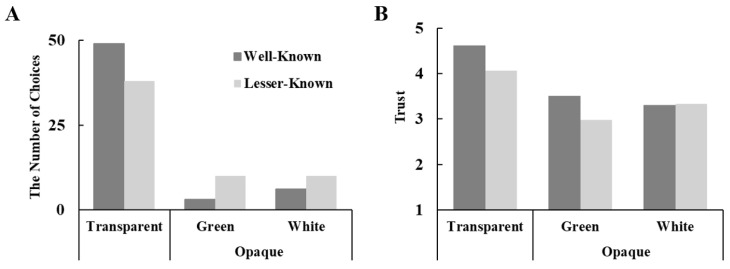
The interaction effect of packaging type (transparent vs. opaque green vs. opaque white) and brand type (well-known vs. lesser-known) on tea preference and trust in Study 3. (**A**) Consumers preferred transparent packaging, an effect strengthened by brand awareness, while opaque green and white showed no difference; (**B**)Transparent packaging increased trust in tea beverages, with a stronger effect for well-known brands and a weaker effect for lesser-known brands.

**Figure 5 foods-14-03893-f005:**
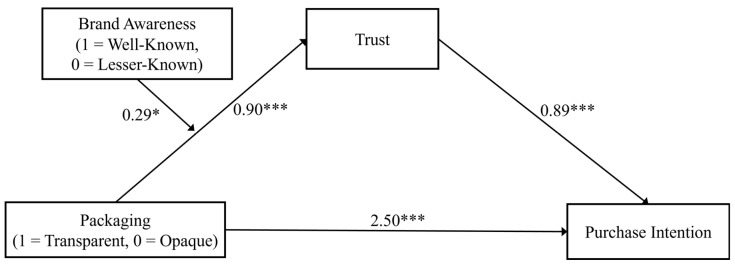
The moderation effect of brand awareness on the mediation effect of trust in Study 3. Note: * indicates *p* < 0.05, *** indicated *p* < 0.001.

## Data Availability

The original contributions presented in the study are included in the article/Appendix A. Further inquiries can be directed to the corresponding authors.

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
