# Peer review of "Transparent Packaging for Tea: Exploring the Role of Trust in Emerging Markets for Tea Beverages"

_foods, 2025, doi:10.3390/foods14223893_

Round 1

Reviewer 1 Report

Comments and Suggestions for Authors

This study investigates the psychological and perceptual effects of transparent vs. opaque packaging on consumer trust and purchase intention in ready-to-drink (RTD) tea beverages. It applies trust theory and crossmodal research frameworks across three experimental studies to test mediation (trust) and moderation (brand awareness) effects. The manuscript is well-structured and contributes meaningfully to the intersection of food marketing, sensory perception, and consumer behavior, particularly in the fast-growing functional beverage market. Detailed comments are listed below:

  1. Describe briefly the types of tea and their properties. Refer to https://doi.org/10.1016/j.chemosphere.2024.143550
  2. Trust theory application could be deepened. The manuscript largely treats trust as a unidimensional construct but should differentiate cognitive trust (belief in product quality) from affective trust (emotional security). These dimensions may respond differently to transparency cues.
  3. Expand on crossmodal congruency (colour–flavour associations) using literature from sensory chemistry and perception to connect visual packaging cues to implicit flavor expectations.
  4. Integrate the cue utilization theory—transparency acts as an extrinsic cue that interacts with intrinsic cues (color, clarity, turbidity) affecting perceived authenticity.
  5. The study uses static digital images; actual product materials (PET, glass) and optical properties could alter consumer response. Future research should specify light transmission and reflectance properties of the packaging materials.
  6. Only binary measures (“transparent” vs. “opaque”) were reported. Including a perceived transparency scale (Likert-based) would provide a more nuanced validation.
  7. Statistical reporting (Add effect sizes, confidence intervals, and power calculations for each dependent variable; clarify how the PROCESS Model 7 analyses were validated: bootstrap sample size, confidence intervals, and significance of indirect effects).
  8. The interaction between brand awareness and transparency requires a stronger theoretical rationale: why does transparency amplify trust for well-known rather than lesser-known brands, contrary to some prior research?
  9. The discussion could connect visual access to chemical trust (i.e., belief in product purity, absence of additives, or oxidation), integrating chemistry-based arguments for perceived authenticity.
  10. The authors effectively demonstrate that transparency increases trust and purchase intention across tea types and brands. However, the discussion should integrate consumer sensory expectations with packaging chemistry—for example, transparency may imply freshness but can also lead to light-induced oxidation in polyphenol-rich teas.
  11. Include a short note on the photochemical risks of transparent packaging (e.g., catechin degradation, chlorophyll photooxidation), which could inform a balanced marketing-science strategy.

Author Response

This study investigates the psychological and perceptual effects of transparent vs. opaque packaging on consumer trust and purchase intention in ready-to-drink (RTD) tea beverages. It applies trust theory and crossmodal research frameworks across three experimental studies to test mediation (trust) and moderation (brand awareness) effects. The manuscript is well-structured and contributes meaningfully to the intersection of food marketing, sensory perception, and consumer behavior, particularly in the fast-growing functional beverage market. Detailed comments are listed below:

Comments 1: Describe briefly the types of tea and their properties. Refer to https://doi.org/10.1016/j.chemosphere.2024.143550

Response 1: Thank you for the suggestion. Based on KaczyÅ„ski et al. (2024), we have now added a brief description at the start of the first paragraph of the Introduction (see Page 4, lines 64–68), outlining the main types of tea (green, white, yellow, oolong, black, and dark teas) and their health-promoting function.

Comments 2: Trust theory application could be deepened. The manuscript largely treats trust as a unidimensional construct but should differentiate cognitive trust (belief in product quality) from affective trust (emotional security). These dimensions may respond differently to transparency cues.

Response 2: Thank you for your comment. We have clarified in the Literature Review section that trust is a multidimensional construct, including cognitive trust, which reflects beliefs in product quality, and affective trust, which reflects emotional security (see Page 7, line 150; Page 8, lines 151–153).

Comments 3: Expand on crossmodal congruency (colour–flavour associations) using literature from sensory chemistry and perception to connect visual packaging cues to implicit flavor expectations.

Response 3: In the revised manuscript (see Page 6, lines 111–116), we expand our discussion of colour–flavour crossmodal correspondences by incorporating the relevant literature from sensory perception and chemistry.

Comments 4: Integrate the cue utilization theory—transparency acts as an extrinsic cue that interacts with intrinsic cues (color, clarity, turbidity) affecting perceived authenticity.

Response 4: We have integrated the cue utilization theory in both the Introduction (see Page 5, lines 99–104) and General Discussion (see Page 23, lines 473–478) in order to clarify the role of packaging transparency.

Comments 5: The study uses static digital images; actual product materials (PET, glass) and optical properties could alter consumer response. Future research should specify light transmission and reflectance properties of the packaging materials.

Response 5: We acknowledge that using static digital images may not fully capture the visual and material properties of real packaging, such as PET or glass. As noted in the Limitations section (see Page 27, lines 577–579; Page 28, lines 583–587), factors including packaging materials and surface finish may also influence both perceived transparency and product stability.

Comments 6: Only binary measures (“transparent” vs. “opaque”) were reported. Including a perceived transparency scale (Likert-based) would provide a more nuanced validation.

Response 6: We agree that using a binary measure may limit the sensitivity of our findings. Accordingly, we discuss this as a potential limitation (see Page 28, lines 596–601).

Comments 7: Statistical reporting (Add effect sizes, confidence intervals, and power calculations for each dependent variable; clarify how the PROCESS Model 7 analyses were validated: bootstrap sample size, confidence intervals, and significance of indirect effects).

Response 7: We have added the effect sizes in the manuscript (see Page 13, lines 262–266). For Studies 2 and 3, we have also reported the p-values for the indirect effects in the moderated mediation analyses (see Page 17, lines 352–355; Page 20, lines 419–423).

Comments 8: The interaction between brand awareness and transparency requires a stronger theoretical rationale: why does transparency amplify trust for well-known rather than lesser-known brands, contrary to some prior research?

Response 8: We adopted trust transfer theory to explain the results. Trust in a retailer or producer can extend to the products or services they offer, ultimately influencing consumer decisions. For high-awareness brands, consumers already hold a baseline level of trust due to prior experience or brand reputation. Transparent packaging thus reinforces and amplifies this pre-existing trust, enhancing perceptions of reliability, quality, and safety. In contrast, lesser-known brands lack an established foundation of trust, limiting the potential for trust transfer; consequently, transparency exerts a weaker effect on consumer trust. We have incorporated this explanation into the revised manuscript (see Page 24, lines 506–515; Page 25, lines 516–517) to clarify the underlying mechanism of the interaction between brand awareness and packaging transparency.

Comments 9: The discussion could connect visual access to chemical trust (i.e., belief in product purity, absence of additives, or oxidation), integrating chemistry-based arguments for perceived authenticity.

Response 9: We have revised the General Discussion to incorporate this point (see Page 23, lines 491–493; Page 24, line 494).

Comments 10: The authors effectively demonstrate that transparency increases trust and purchase intention across tea types and brands. However, the discussion should integrate consumer sensory expectations with packaging chemistry—for example, transparency may imply freshness but can also lead to light-induced oxidation in polyphenol-rich teas.

Response 10: We have revised the Limitations to note that transparent packaging can be made from different materials, such as PET or glass, and highlighted the potential oxidation risks these materials may pose for tea beverages (see Page 27, lines 577–579; Page 28, lines 583–587).

Comments 11: Include a short note on the photochemical risks of transparent packaging (e.g., catechin degradation, chlorophyll photooxidation), which could inform a balanced marketing-science strategy.

Response 11: As suggested, we briefly note the photochemical risks of transparent packaging in the Limitations section of the revised manuscript (see Page 27, lines 580–581; Page, lines 582–583).

Reviewer 2 Report

Comments and Suggestions for Authors

This study needs some changes before considered for acceptance

1. There is an ambiguity in the introduction section, the authors have placed the health benefits of tea beverages/RTD tea at par with the conventional brewed tea. Is this statement correct?

2. What was the exclusion criteria of other participants which were reduced from 194-152 in study 1?

3. Souldn't purchase intention include any phychological parameter as it is based on trust rather healthiness and tastiness are measured.

4. Was the transparent packaging used as primary packaging or secondary packaging, as if it used as primary packaging then what is the impact of lightness or darkness impact on tea 

5. There is no specific exclusion criteria mentioned as there are 147 participants in tastiness and different number of participants in purchase intention and healthiness

6. Mentioned results are of opaque red and green in study 2 while procedure mentions opaque white, green, and yellow.

Author Response

This study needs some changes before considered for acceptance

Comments1: There is an ambiguity in the introduction section, the authors have placed the health benefits of tea beverages/RTD tea at par with the conventional brewed tea. Is this statement correct?

Response 1: We thank the reviewer for this comment. Although RTD tea beverages are processed and sold in liquid form, they are still prepared through the extraction of tea leaves. According to Liang et al. (2021), modern advanced blending techniques largely preserve the flavor and quality derived from the raw tea materials. However, as acknowledged in the Introduction, RTD teas may be perceived by consumers as less healthy than their traditionally brewed counterparts, primarily due to the inclusion of added ingredients, such as sugar. We have clarified this point in the Introduction section to avoid potential ambiguity (see Page 4, lines 63–64, 70–72).

Comments 2: What was the exclusion criteria of other participants which were reduced from 194-152 in study 1? 

Response 2: In Study 1, the number 152 represents the minimum required sample size estimated using G*Power based on prior research on packaging transparency, rather than the final sample size. We initially recruited 200 participants and excluded six who failed the manipulation check (i.e., could not correctly distinguish between transparent and opaque packaging), resulting in a final valid sample of 194 participants. We have reorganized the description of the sample size calculation to prevent any potential misunderstanding regarding the actual sample size in Study 1 (see Page 10, lines 211–215).

Comments 3: Souldn’t purchase intention include any phychological parameter as it is based on trust rather healthiness and tastiness are measured.

Response 3: We have added this as a limitation and suggested that future research could incorporate additional psychological or behavioral measures (see Page 27, lines 572–577).

Comments 4: Was the transparent packaging used as primary packaging or secondary packaging, as if it used as primary packaging then what is the impact of lightness or darkness impact on tea 

Response 4: We have noted this as a limitation and suggested that future research could examine how the optical properties of transparent packaging, such as light transmission, influence both tea quality and consumer evaluations (see Page 27, lines 577–581; Page 28, lines 582–587).

Comments 5: There is no specific exclusion criteria mentioned as there are 147 participants for tastiness and different number of participants for purchase intention and healthiness

Response 5: The final valid sample in Study 1 consisted of 194 participants, with 44 in the transparent packaging group and 50 in each of the three opaque packaging groups (green, yellow, and white). The number 147 in the F statistic represents the error degrees of freedom for the ANOVA comparing the three opaque packaging groups. We have added this in the Results section (see Page 12, lines 251–253).

Comments 6: Mentioned results are of opaque red and green in study 2 while procedure mentions opaque white, green, and yellow.

Response 6: We have carefully checked the colour descriptions and corrected the inconsistency (see Page 14, lines 289–291).

Reviewer 3 Report

Comments and Suggestions for Authors

Dear authors,

I would like to congratulate you on your work. In order to improve it, I advise you to address the following:

-period of the study (summer, winter..).

-nationality of the respondents. Chinese I presume, but please mention that in the paper.

-tea is usually served hot. Bottled ready-to-drink tea or tea based beverage is usually consumed cold. I advise you to address this in the relevant paper sections.

-study 2, line 22. You mention red packaging while in the section 4.2 (line 241) you state yellow colour. Please clarify this.

-Apart from sensory perception, brand name, healthiness, etc., price also affects purchase intentions. However, it is not mentioned throughout the paper. I suggest you to revise the paper in this regard.

-I strongly advise you to include pictures in the paper, at least in the Appendix.

Author Response

Dear authors,

I would like to congratulate you on your work. In order to improve it, I advise you to address the following:

Comments 1: -period of the study (summer, winter..).

Response 1: The data collection periods for the three studies were December 2024 (see Page 10, lines 213–215), January 2025 (see Page 14, lines 283–285), and April 2025 (see Page 18, lines 381–382), respectively. We have included these time points in the Methods sections of all three studies to provide a clearer context for the research.

Comments 2: -nationality of the respondents. Chinese I presume, but please mention that in the paper.

Response 2: We have clarified the nationality of the respondents, indicating that all participants were indeed Chinese (see Page 10, lines 213–215; Page 14, lines 283–285; Page 18, lines 381–382).

Comments 3: -tea is usually served hot. Bottled ready-to-drink tea or tea based beverage is usually consumed cold. I advise you to address this in the relevant paper sections.

Response 3:  We appreciate the reviewer’s comment regarding the typical serving temperature of tea. While tea is usually served hot, cold-brewed tea has also long been popular. According to Muller et al. (2020) and Yang et al. (2025), many teas, such as rooibos and green tea, are commonly enjoyed in cold-brewed form. Similarly, ready-to-drink tea beverages are typically sold as liquids at ambient temperature or refrigerated. We have clarified this point in the Introduction section of the manuscript (see Page 4, lines 76–79).

Comments 4: -study 2, line 22. You mention red packaging while in the section 4.2 (line 241) you state yellow colour. Please clarify this.

Response 4: We thank the reviewer for pointing out the inconsistency in the packaging colour in Study 2. We have corrected this error in the manuscript (see Page 14, lines 289–291).

Comments 5: -Apart from sensory perception, brand name, healthiness, etc., price also affects purchase intentions. However, it is not mentioned throughout the paper. I suggest you to revise the paper in this regard.

Response 5:  We thank the reviewer for pointing out the potential role of price in influencing purchase intentions. We have discussed it in the manuscript (see Page 25, lines 523–525) as an additional factor that may shape consumer choices.

Comments 6: -I strongly advise you to include pictures in the paper, at least in the Appendix.

Response 6:  We have added images of the tea beverages with different packaging in the Appendix (See details in Appendix 2 and 3) to help readers better understand the stimulus used in the studies.

Reviewer 4 Report

Comments and Suggestions for Authors
  1. General Assessment

This manuscript investigates the influence of packaging transparency on consumers’ trust and purchase intention for ready-to-drink tea beverages. Three experimental studies provide convergent evidence that transparent packaging enhances perceived healthiness, trust, and purchase intention, with brand awareness moderating this relationship. The study is timely and relevant for both academic and industry audiences. Overall, the manuscript is well-structured and methodologically competent, though several theoretical and analytical aspects warrant further refinement.

  1. Comments
  2. Clarify whether the trust construct is cognitive, affective, or both. Current treatment is unidimensional; this limits conceptual precision.
  3. Figure 1 should be explicitly linked to each hypothesis (H1–H3). Describe each path’s theoretical rationale more clearly.
  4. (Study 3) The PROCESS output shows an adverse indirect effect, yet the text interprets it positively. Please clarify which direction the moderation operates—does transparency work better for high- or low-awareness brands?
  5. I suggest discussing practical significance, not only statistical significance.
  6. Since participants were all from China, discuss how cultural colour symbolism (e.g., red = luck) may affect generalizability.
  7. Condense repetitive results and emphasise new insights in the Discussion.
  8. Expand the Limitations section to include light exposure and oxidation issues for transparent packaging.

Author Response

This manuscript investigates the influence of packaging transparency on consumers’ trust and purchase intention for ready-to-drink tea beverages. Three experimental studies provide convergent evidence that transparent packaging enhances perceived healthiness, trust, and purchase intention, with brand awareness moderating this relationship. The study is timely and relevant for both academic and industry audiences. Overall, the manuscript is well-structured and methodologically competent, though several theoretical and analytical aspects warrant further refinement.

Comments 1: Clarify whether the trust construct is cognitive, affective, or both. Current treatment is unidimensional; this limits conceptual precision.

Response 1:  We appreciate this insightful comment. In this research, trust was measured primarily at the cognitive level, reflecting consumers’ beliefs in product quality and reliability (See details in Appendix 1). We have clarified in the Literature Review that trust is a multidimensional construct, including cognitive trust, and affective trust, which reflects emotional security (see Page 7, line 150; Page 8, lines 151–153).

Comments 2: Figure 1 should be explicitly linked to each hypothesis (H1–H3). Describe each path’s theoretical rationale more clearly.

Response 2: In the revised manuscript (see Page 10, lines 200–202), Figure 1 has been updated to explicitly link each path to the corresponding hypothesis (H1–H3).

Comments 3: (Study 3) The PROCESS output shows an adverse indirect effect, yet the text interprets it positively. Please clarify which direction the moderation operates—does transparency work better for high- or low-awareness brands?

Response 3: We have corrected the labeling errors in both the figure and the related text (see Page 20, lines 419–426). The results confirm that transparency exerts a stronger positive effect on well-known brands compared with lesser-known brands.

Comments 4: I suggest discussing practical significance, not only statistical significance.

Response 4: We have revised the Discussion sections for each study (see Page 13, lines 266–269; Page 17, lines 357–363, 366–367; Page 21, lines 448–449; Page 22, lines 450–451).

Comments 5: Since participants were all from China, discuss how cultural colour symbolism (e.g., red = luck) may affect generalizability.

Response 5:  We have reorganized the relevant sections in the General Discussion to further clarify the role of cultural colour symbolism (see Page 23, lines 482–484).

Comments 6: Condense repetitive results and emphasise new insights in the Discussion.

Response 6: We have reorganized the Discussion sections for all three studies (see Page 13, lines 259–269; Page 17, lines 357–363, 366–367; Page 21, lines 448–449; Page 22, lines 450–451). Repetitive presentation of results has been condensed.

Comments 7: Expand the Limitations section to include light exposure and oxidation issues for transparent packaging.

Response 7: We appreciate the reviewer’s suggestion. We have expanded the Limitations section (see Page 27, lines 577–581; Page 28, lines 582–587) to note that transparent packaging may expose tea beverages to light, potentially leading to oxidation and degradation of key compounds. We suggest that future research should more thoroughly investigate the oxidation risks associated with transparent packaging, particularly for light-sensitive products.

Reviewer 5 Report

Comments and Suggestions for Authors

The effect of transparent vs. opaque packaging has already been studied, as the authors themselves report. Additionally, the influence of colour on perceived health, taste, and similar attributes has been previously researched and reported, and a review paper on this topic is available. However, it should be noted that the aforementioned analyses do not include tea beverages, and in this respect, the present work offers a certain novelty and complements the field that has already been partially explored.

 I recommend including the images used in the experiment. Images could be added to supplementary material where the questions are provided, because without these images, it is difficult to fully interpret the results. The work is clearly and consistently structured, the results are appropriately presented and discussed, and the relevant literature is addressed. The conclusions are consistent and follow logically from the results.

Please check line 241: the colours do not match.

Lines 313–315: clarify the statement that this packaging is better from this perspective.

Author Response

The effect of transparent vs. opaque packaging has already been studied, as the authors themselves report. Additionally, the influence of colour on perceived health, taste, and similar attributes has been previously researched and reported, and a review paper on this topic is available. However, it should be noted that the aforementioned analyses do not include tea beverages, and in this respect, the present work offers a certain novelty and complements the field that has already been partially explored.

Comments 1: I recommend including the images used in the experiment. Images could be added to supplementary material where the questions are provided, because without these images, it is difficult to fully interpret the results. The work is clearly and consistently structured, the results are appropriately presented and discussed, and the relevant literature is addressed. The conclusions are consistent and follow logically from the results.

Response 1:  We have now included the images used in the experiment as supplementary material (see Appendix 2 and 3).

Comments 2: Please check line 241: the colours do not match.

Response 2:

Thank you for noticing this. We have carefully checked the colour descriptions and corrected the inconsistency to ensure that the reported colours now accurately match the corresponding stimuli (see Page 14, lines 289–291).

Comments 3: Lines 313–315: clarify the statement that this packaging is better from this perspective.

Response 3: We appreciate the reviewer’s suggestion. We have revised the statement (see Page 17, lines 366–367; Page 18, line 368) to clarify that the superiority of transparent packaging specifically refers to its effect on enhancing consumers’ trust.

Round 2

Reviewer 1 Report

Comments and Suggestions for Authors

The Authors have improved the paper. I have no more comments.

Reviewer 2 Report

Comments and Suggestions for Authors

The authors have addressed the revision points in a good manner and article is good enough to be proceeded 

Reviewer 5 Report

Comments and Suggestions for Authors

The authors have taken the suggestions into account. Most importantly, the images they have added make it much easier to interpret and understand the results. In my opinion, the paper in its current form can be accepted for publication.